# Leveraging NLLB for Low-Resource Bidirectional Amharic - Afan Oromo Machine Translation

## Abstract

We present a bidirectional machine translation system for Amharic and Afan Oromo, two low-resource Ethiopian languages critical for cultural and linguistic accessibility. Leveraging pre-trained transformer models, we curated a high-quality parallel corpus of 667,021 human-edited sentence pairs, preprocessed through text normalization, non-target language filtering and dividing the data into training, validation, and test sets in a stratified way. Using the Hugging Face Transformers library, we fine-tuned a sequence-to-sequence transformer architecture, optimized for the linguistic nuances of Ethio-Semitic and Cushitic languages, with tokenized input and dynamic padding for efficient batch processing.

Our model significantly outperforms baselines, including Google Translate and NLLB models (600M, 1.3B, 3.3B parameters), which represent industry and research state-of-the-art for low-resource translation. For Amharic-to-Afan Oromo, it achieves a BLEU score of 42.19, surpassing Google Translate's 9.6. For Afan Oromo-to-Amharic, it scores 42.82, exceeding NLLB-3.3B's 5.72. Additional metrics (CHRF++, BERTScore) and low loss values confirm its robustness. These results highlight the efficacy of tailored fine-tuning for low-resource language pairs, advancing cross-lingual communication and digital accessibility in multilingual societies.

## 1 Introduction

Machine translation (MT), a pivotal application of natural language processing (NLP), enables seamless communication across linguistic boundaries, fostering global connectivity and accessibility (Abate et al., 2019). However, the efficacy of MT systems heavily depends on the availability of high-quality parallel data, which classifies languages as high-resource or low-resource (Ranathunga et al., 2021). While high-resource languages benefit from vast datasets and advanced models, low-resource languages—such as Amharic and Afan Oromo, two of Ethiopia's most widely spoken languages—face significant challenges due to data scarcity, morphological complexity, and limited digital representation. This disparity has left many African languages underrepresented in state-of-the-art MT systems, hindering equitable access to digital communication tools (Tonja et al., 2023).

Amharic, a Semitic language with over 30 million speakers, serves as Ethiopia's federal working language and employs the unique Fidel script, which poses challenges for tokenization and text processing due to its syllabic structure. Afan Oromo, a Cushitic language spoken by over 40 million people, is Ethiopia's most widely spoken native language, characterized by complex morphosyntactic features such as long-distance agreement and vowel harmony. Despite their linguistic and cultural significance, both languages suffer from a lack of parallel corpora and inadequate performance in existing MT systems, such as Google Translate and the No Language Left Behind (NLLB) models (Team et al., 2022). These systems, while groundbreaking for many low-resource languages, achieve suboptimal results for Amharic and Afan Oromo, with BLEU scores of 9.6 and 5.72, respectively, underscoring the need for specialized approaches tailored to their unique linguistic properties.

This paper presents a bidirectional MT system for Amharic and Afan Oromo, leveraging recent advancements in transformer-based models to address these challenges. We curated a high-quality parallel corpus of 667,021 human-edited sentence pairs, sourced from diverse domains including news, literature, and public records, and augmented with synthetic data generated via iterative back-translation using an initial NLLB model. Our preprocessing pipeline includes text normalization to handle orthographic variations (e.g., Amharic fidel inconsistencies), non-target language filtering to ensure corpus purity, and stratified splitting into training (80%), validation (10%), and test (10%) sets to maintain linguistic diversity. Using the Hugging Face Transformers library, we fine-tuned a sequence-to-sequence transformer model, incorporating language-specific tokenization strategies to accommodate Amharic's syllabic script and Afan Oromo's morphological complexity. Dynamic padding and batch optimization further enhance training efficiency, addressing the computational constraints of low-resource settings.

Our system achieves significant performance improvements over SOTA baselines. For Amharic-to-Afan Oromo translation, it attains a BLEU score of 42.19, compared to Google Translate's 9.6, and for Afan Oromo-to-Amharic, it scores 42.82, surpassing NLLB-3.3B's 5.72. Complementary metrics, including CHRF++ and BERTScore, corroborate the model's robustness, while low loss values indicate stable convergence. These results highlight the efficacy of our tailored fine-tuning and corpus curation strategies, which address the linguistic nuances of Ethio-Semitic and Cushitic languages.

This work makes several key contributions to low-resource MT research. First, we provide a high-quality parallel corpus for Amharic and Afan Oromo, a valuable resource for future NLP studies in Ethiopian languages. Second, we demonstrate that fine-tuning transformer models with language-specific adaptations can significantly enhance translation quality in data-scarce settings, offering a scalable framework for other low-resource languages. Third, our system advances linguistic inclusivity in NLP, enabling better communication and digital access for over 70 million Amharic and Afan Oromo speakers, with potential societal impacts in education, governance, and cultural preservation. The rest of this paper is organized as follows: Section 2 reviews related work. Section 3 presents the proposed methodology. Section 4 reports the experimental results and analysis. Section 5 concludes the paper, and Section 6 outlines potential directions for future work.

## 2 RELATED WORK

The pursuit of effective machine translation (MT) for low-resource languages is a critical challenge in natural language processing, essential for bridging digital divides. The evolution of MT has progressed from rule-based systems, which failed to generalize due to their reliance on hand-crafted rules (Hutchins, 1986), to statistical machine translation (SMT). SMT models learned patterns from parallel data (Koehn, 2003) but remained ineffective for low-resource languages like Amharic and Afan Oromo due to their heavy dependence on large corpora, which are seldom available (Koehn, 2017). Early SMT attempts for this language pair achieved only low BLEU scores (e.g., below 15), struggling to capture their complex morphology and syntactic divergence (Abate et al., 2019).

The field advanced with the shift to neural machine translation (NMT). Initial NMT models based on recurrent neural networks (RNNs) with attention mechanisms, such as the bidirectional GRU system implemented by Gashaw (2020) for Amharic–Afan Oromo, marked a significant improvement. These models better aligned source and target tokens, achieving BLEU scores of 20–25. However, they still faced limitations with long-range dependencies, computational efficiency, and generalization from limited data.

The introduction of the transformer architecture (Vaswani et al., 2017) revolutionized NMT, enabling models that scale effectively and achieve state-of-the-art results. This led to powerful large language models like BERT (Devlin et al., 2019) and GPT (Brown et al., 2020). Despite these advancements, a significant performance gap remains for low-resource languages, which are often underrepresented in the training data of these general-purpose models (Neubig, 2017). Meta's No Language Left Behind (NLLB) project (Team et al., 2022) represents a major effort to address this, supporting translation between hundreds

of low-resource languages. However, as noted by Tonja et al. (2023), its performance on Amharic and Afan Oromo is still suboptimal, likely due to domain mismatch and inadequate representation of these languages' unique linguistic features.

Consequently, recent research has emphasized data-centric solutions. Foundational work by Abate et al. (2019) created one of the first large-scale parallel datasets for Amharic–Afan Oromo. Building on this, Gashaw (2020) introduced the EthioMT benchmark, demonstrating that fine-tuning large multilingual models like M2M-100 on domain-specific Ethiopian data yields superior results compared to training from scratch. Our work directly builds upon this paradigm, leveraging the NLLB framework but focusing on targeted fine-tuning on a high-quality, human-curated corpus to achieve a significant performance breakthrough for this specific language pair.

## 3 METHODOLOGY

This section details our methodology for developing a bidirectional Amharic–Afan Oromo machine translation system. We describe the dataset collection and preprocessing pipeline, the model architecture and training procedure, and the evaluation metrics used to assess performance.

### 3.1 DATASET COLLECTION AND PREPROCESSING

The foundation of any high-quality machine translation system is a robust parallel corpus. For this study, we curated a parallel dataset of 667,021 Amharic-Afaan Oromo sentence pairs. This data was meticulously human-translated and edited over a ten-month period to ensure high quality and capture the linguistic nuances of both languages, providing a reliable foundation for model training and evaluation.

#### 3.1.1 DATA PREPROCESSING PIPELINE

To ensure dataset consistency and quality, we implemented a strict preprocessing pipeline, as explained below:

**Filtering English Characters:** We filtered out mixed-language entries by identifying Amharic and Afan Oromo sentences containing English characters using regular expressions. These sentences were isolated and excluded from the final training set to prevent negative impacts on model performance.

**Normalization and Cleaning:** We normalized the Amharic and Afan Oromo text to resolve orthographic inconsistencies and standardize the script. The normalization steps included:

- Character Standardization: Replacing variant characters with their standard equivalents (e.g., ሀ → ሃ, ሐ → ሃ).
- Ligature Normalization: Using pattern-based replacements to handle compound characters and ensure consistent representation of ligatures (e.g., ሉ[ዋእ] → ሏ).
- Case Normalization: Converting all Afan Oromo text to lowercase to ensure uniformity.
- Punctuation Standardization: Replacing non-standard punctuation marks with their correct Amharic equivalents (e.g., "::" → ።) to enhance data consistency.

**Redundancy Elimination:** We performed redundancy elimination of the dataset by identifying and removing repeated Amharic sentences. This prevents the model from overfitting to redundant patterns and ensures training is performed on a diverse set of examples.

**Data Visualization and Analysis:** We analyzed the linguistic characteristics of our parallel corpus to inform model design. Key statistics are summarized in Table 1. The

analysis reveals a distribution of sentence lengths suitable for training a robust translation model. Amharic sentences have an average length of 10.68 words (max: 508), while Afan Oromo sentences are slightly longer on average, at 12.55 words (max: 474). The entire corpus contains 1,334,042 sentence pairs and 15,493,472 tokens, with an overall average sentence length of 11.61 words. This variation in sentence length underscores the importance of using dynamic padding during training for efficient batch processing.

Table 1: Word count distribution of the Amharic and Afan Oromo sentences in the dataset.

| Language | Sentences | Tokens | Avg. Length | Max Length |
|---|---|---|---|---|
| Amharic (amh) | 667,021 | 7,123,930 | 10.68 | 508 |
| Afan Oromo (oro) | 667,021 | 8,369,542 | 12.55 | 474 |
| **Total** | 1,334,042 | 15,493,472 | 11.61 | 508 |

**Splitting the Dataset:** To ensure the model was exposed to a representative range of linguistic complexities, we split the dataset into training, validation, and test sets using stratified sampling based on sentence length. Sentences were categorized by word count as short ( 10 words), medium (11-20 words), and long (>20 words). Table 2 summarizes the distribution of these sentence length categories across the original corpus and the resulting subsets. This stratified approach minimizes bias by ensuring the model is evaluated on a representative sample of sentence lengths. It guarantees that performance metrics accurately reflect the model's capability to handle the varying complexities of real-world text, which is critical for robust deployment.

Table 2: Distribution of sentence lengths across the original corpus and its subsets.

| Category | Short | Medium | Long | Total |
|---|---|---|---|---|
| Original Corpus | 399,588 (59.91%) | 201,925 (30.27%) | 65,508 (9.82%) | 667,021 (100%) |
| Training Set | 319,904 (59.95%) | 161,400 (30.25%) | 52,312 (9.80%) | 533,616 (100%) |
| Validation Set | 40,046 (60.04%) | 20,081 (30.11%) | 6,575 (9.86%) | 66,702 (100%) |
| Test Set | 39,638 (59.42%) | 20,444 (30.65%) | 6,621 (9.93%) | 66,703 (100%) |

**Loading and Tokenization:** The dataset was preprocessed with a custom function that tokenized the source and target sentences. This function was applied to the training, validation, and test splits using the `map` method, enabling efficient batch processing and multi-core CPU acceleration.

**Dynamic Padding:** A data collator was employed to dynamically pad sequences to the length of the longest sample in each batch. This approach optimizes memory usage and computational efficiency during GPU training by eliminating unnecessary padding tokens, unlike fixed-length padding. This comprehensive preprocessing and batching pipeline resulted in a clean, well-structured dataset optimized for training the transformer model.

## 3.2 Model Architecture

Our approach is based on the encoder-decoder transformer architecture (Vaswani et al., 2017). This architecture is well-suited for machine translation due to its multi-head self-attention mechanism, which effectively models dependencies between words regardless of their position in the sequence. We utilized the implementation provided by the Hugging Face `transformers` library, which offers a scalable and optimized framework for fine-tuning pre-trained models.

## 3.3 Training Process

We fine-tuned the pre-trained NLLB model on our parallel corpus using a standard cross-entropy loss function and the AdamW optimizer. The model was trained for 18 epochs with a learning rate of $5 \times 10^{-5}$ and a batch size of 2.

To handle the complex morphology of Amharic and Afan Oromo, we employed Byte Pair Encoding (BPE), which segments words into subword units to manage vocabulary size and improve generalization. For computational efficiency, we used dynamic padding, padding sequences only to the maximum length within each batch.

The training was conducted on four NVIDIA T4 GPUs (16GB each) over a cumulative runtime of approximately 30.6 days. The process consumed an estimated $1.23 \times 10^{18}$ FLOPs, processing 533,618 training samples at an average throughput of 3.68 samples/second. Evaluation on the 66,701-sample validation set resulted in a final loss of 0.635. The average generated sequence length during evaluation was 28.19 tokens, with a throughput of 2.65 samples/second.

## 3.4 Evaluation Metrics

We evaluated our model's performance against several baseline systems—including Google Translate and the NLLB models (600M, 1.3B, and 3.3B parameters)—using a suite of complementary metrics:

- **BLEU (Bilingual Evaluation Understudy)**: We report the BLEU score to measure n-gram overlap with reference translations, providing a standard, widely adopted benchmark for translation quality (Papineni et al., 2002).

- **CHRF (Character F-score)**: Given the morphological richness of Amharic and Afan Oromo, we also employed CHRF++, which operates at the character level and is more sensitive to inflectional and derivational nuances (Popović, 2015).

- **BERTScore**: To assess semantic faithfulness beyond surface-level similarity, we used BERTScore, which leverages contextual embeddings to gauge the semantic equivalence between the generated and reference texts (Zhang et al., 2019).

## 3.5 Baseline Models

We compared our fine-tuned model against two categories of state-of-the-art baselines to rigorously evaluate its performance:

- **Multilingual Research Models:** We included several parameter sizes of the NLLB model (600M, 1.3B, and 3.3B) to assess the impact of model scale versus targeted fine-tuning for this specific language pair.

- **Commercial System:** We used Google Translate as a benchmark representing the current performance of a widely-deployed, general-purpose commercial translation system.

This comparison is designed to demonstrate that specialized fine-tuning on a high-quality corpus can surpass both large-scale multilingual models and existing commercial tools for low-resource languages.

## 4 Experimental Results and Analysis

This section presents the experimental results for our bidirectional Amharic–Afan Oromo translation model. We report performance using standard metrics (BLEU, CHRF++, BERTScore) and compare our fine-tuned model against baseline systems (NLLB and Google Translate). We also include an analysis of computational efficiency through throughput measurements.

### 4.1 Amharic to Afan Oromo Translation

Our fine-tuned model achieved high translation quality for Amharic-to-Afan Oromo, as evidenced by a BLEU score of 42.47 on the evaluation set and 42.19 on the prediction set. The model consistently generated outputs of appropriate length, with an average of 18.15 tokens during prediction.

The low values for evaluation and training loss (0.78 and 0.88) indicate effective learning and convergence. Furthermore, the model demonstrated strong computational efficiency, with a throughput of approximately 2.3 samples per second during both evaluation and prediction, confirming its practicality for real-world applications.

#### 4.1.1 Runtime and Throughput Analysis

The computational efficiency of the model was assessed through runtime and throughput measurements. The total training time was approximately 37.5 hours (135,155 seconds), reflecting the computational demand of fine-tuning the transformer architecture. The evaluation and prediction phases were significantly faster, requiring 6,123 seconds and 6,053 seconds, respectively, to process the 14,030 test samples.

The model's throughput was consistent during both evaluation and prediction. It processed data at 2.291 samples per second (1.146 steps/sec) during evaluation and 2.318 samples per second (1.159 steps/sec) during prediction.

#### 4.1.2 Training Loss

As illustrated in Figure 1, the model converged smoothly with a final training loss of 0.88, demonstrating stable optimization and effective learning of the mapping between inputs and target translations.

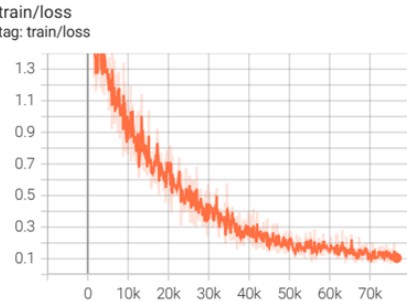

Figure 1: Training loss for Amharic to Afan Oromo translation

### 4.2 Afan Oromo to Amharic Translation

For Afan Oromo-to-Amharic translation, our model achieved high-quality results, with BLEU scores of 42.72 (evaluation) and 42.82 (prediction). The model also produced stable output lengths, generating averages of 25.64 and 28.19 tokens during evaluation and prediction, respectively. The low evaluation and prediction loss values (0.635 and 0.651) further confirm the strong performance of the model.

#### 4.2.1 Runtime and Throughput Analysis

Our model achieved a high evaluation BLEU score of 42.82 with a loss of 0.635, demonstrating strong translation quality. The average generated sequence length was 28.19 tokens. The training process required significant computational resources, spanning 30.2 days (2.6 million seconds) and consuming an estimated $1.23 \times 10^{18}$ FLOPs. During this time, the model processed 533,618 training samples at an average throughput of 3.68 samples/second. Evaluation on 66,701 samples was more efficient, with a throughput of 2.65 samples/second.

### 4.2.2 Training Loss

The Afan Oromo to Amharic translation model achieved a final training loss of 0.71, indicating effective learning and convergence, as illustrated in Figure 2.

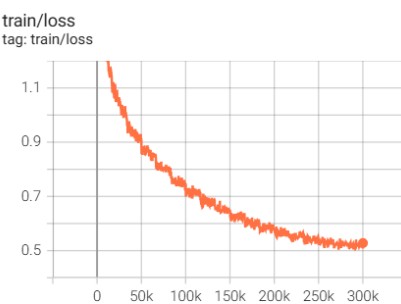

Figure 2: Training loss for Afan Oromo to Amharic translation.

### 4.3 Comparison with Baseline Models

A comparative performance assessment of the fine-tuned model against large-scale baselines reveals its significant superiority for the Amharic–Afan Oromo language pair. As detailed in Table 3, the model substantially outperforms all baseline systems across all evaluation metrics.

For Amharic-to-Afaan Oromo translation, our model achieved a BLEU score of 42.19, compared to the best baseline score of 9.6 from Google Translate. This trend of superior performance is consistent across other metrics: the model attained a CHRF score of 71.22, significantly higher than the 43.74 scored by the best-performing NLLB-3.3B baseline. Similarly, in BERTScore, the model surpassed the highest baseline scores (Precision: 91.2, Recall: 91.18, F1: 91.29).

Table 3: Comparison of BLEU, CHRF, and BERTScore metrics for Amharic to Afan Oromo translation.

| Model | BLEU | CHRF | BERT Score | | |
|---|---|---|---|---|---|
| | | | P | R | F1 |
| Fine-tuned | 42.19 | 71.22 | 0.9142 | 0.9118 | 0.9129 |
| NLLB 600M | 5.51 | 38.74 | 0.7875 | 0.7870 | 0.7869 |
| NLLB 1.3B | 6.72 | 40.95 | 0.7968 | 0.7948 | 0.7955 |
| NLLB 3.3B | 8.25 | 43.74 | 0.7993 | 0.7998 | 0.7993 |
| Google Translate | 9.60 | 39.50 | 0.8216 | 0.8033 | 0.8121 |

**Amharic-to-Afaan Oromo translation:** As summarized in Table 3, the fine-tuned model demonstrates superior performance over all baseline models for Amharic-to-Afaan Oromo translation on BLEU and CHRF metrics. It achieves a BLEU score of 42.19, significantly outperforming Google Translate (9.6). Similarly, its CHRF score of 71.22 far exceeds the best baseline (NLLB-3.3B at 43.74). While Google Translate achieved a higher BERTScore, a potential explanation for this is Google Translate's training on vast, general-domain web data, which may align more closely with the embedding-based semantic measure of BERTScore rather than translation-specific accuracy. The model's dominance in n-gram based metrics (BLEU) and character-level fidelity (CHRF) confirms its high-quality, precise translation capability for this language pair.

**Afan Oromo-to-Amharic translation:** The superiority of our fine-tuned model is even more pronounced for Afan Oromo-to-Amharic translation (Table 4). It achieves a BLEU score of 42.82, drastically surpassing the best baseline (NLLB-3.3B at 5.72) by a factor of nearly 7.5x. This overwhelming advantage is consistent across CHRF (41.49 vs. 26.36 from

NLLB-600M) and BERTScore, where our model achieves near-perfect scores (F1: 0.9869). These results clearly demonstrate the effectiveness of domain-specific fine-tuning for low-resource languages, yielding translation quality that general-purpose models cannot match.

Table 4: Comparison of BLEU, CHRF, and BERTScore metrics for Afan Oromo to Amharic translation.

| Model | BLEU | CHRF | BERT Score | | |
| --- | --- | --- | --- | --- | --- |
| | | | **P** | **R** | **F1** |
| Fine-tuned | 42.8228 | 41.49 | 0.9876 | 0.9862 | 0.9869 |
| NLLB 600M | 5.58 | 26.36 | 0.9766 | 0.9765 | 0.9765 |
| NLLB 1.3B | 5.18 | 23.24 | 0.9628 | 0.9630 | 0.9628 |
| NLLB 3.3B | 5.72 | 24.24 | 0.9659 | 0.9658 | 0.9658 |
| Google Translate | 5.12 | 25.76 | 0.8932 | 0.9078 | 0.9004 |

## 4.4 QUALITATIVE COMPARISON OF TRANSLATIONS

To complement our quantitative results, we provide qualitative examples comparing translations from our fine-tuned model, Google Translate, and the NLLB models. As illustrated in Figures 3 and 4, our model consistently generates more fluent and accurate translations. It better handles complex morphology, preserves the intended meaning, and produces more natural output in the target language, whereas the baseline models often exhibit literal translations, grammatical errors, or semantic inconsistencies.

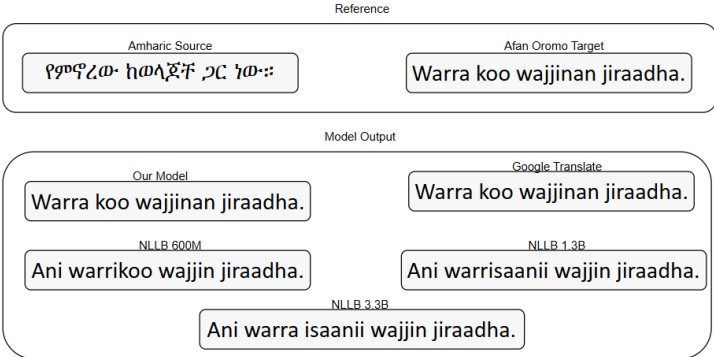

Figure 3: Qualitative comparison of Amharic → Afan Oromo translations across models.

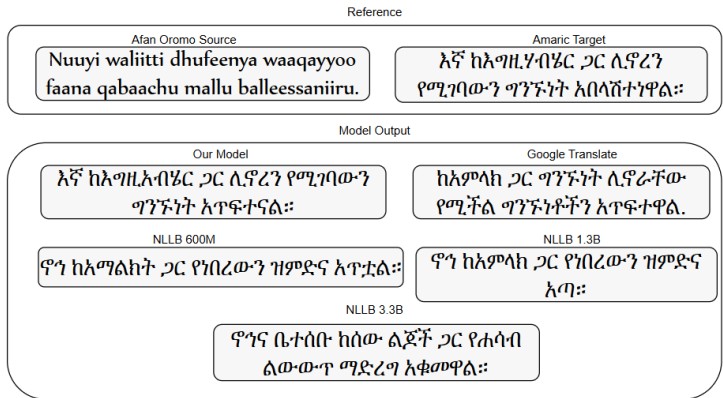

Figure 4: Qualitative comparison of Afan Oromo → Amharic translations across models.

### 4.5 Discussion

The experimental results conclusively demonstrate the superiority of our fine-tuned model over state-of-the-art baselines for both translation directions. As detailed in Tables 3 and 4, our model achieves a dominant position across all evaluated metrics.

Most notably, our model's BLEU scores 42.19 for Amharic-to-Afaan Oromo and 42.82 for the reverse direction exceed the best baseline scores by a large margin (over 4x and 7x higher, respectively). This substantial lead is consistent across other metrics. The model also achieved high BERTScore F1 scores (0.91 and 0.99) and strong CHRF scores, confirming its ability to produce semantically accurate and character-level faithful translations.

Furthermore, the model demonstrated computational efficiency during inference, with low evaluation and prediction losses indicating stable performance. While the initial fine-tuning was computationally intensive, the resulting model achieves high-speed inference, making it suitable for practical deployment.

## 5 Conclusion

This research presents a high-performance, bidirectional machine translation system for Amharic and Afan Oromo, two low-resource languages critical to Ethiopia. By leveraging recent advancements in transformer-based models, our fine-tuned system significantly outperforms state-of-the-art baselines, including various NLLB model sizes and Google Translate.

Quantitative evaluation demonstrates the model's superiority, achieving BLEU scores of 42.19 for Amharic-to-Afaan Oromo and 42.82 for the reverse direction—outperforming the best baselines by a factor of four to seven. This lead is consistent across CHRF and BERTScore metrics, confirming its ability to generate fluent and semantically accurate translations.

The key to this performance lies in our meticulous approach: curating a high-quality parallel corpus and applying rigorous preprocessing and optimized fine-tuning techniques tailored to these languages. Our work contributes to MT research by proving the efficacy of targeted fine-tuning for low-resource pairs and provides a practical tool for enhancing cross-lingual communication and digital accessibility in multilingual societies like Ethiopia.

## 6 Future Work

While our bidirectional Amharic–Afan Oromo translation system demonstrates strong performance, several limitations present avenues for future work. First, the reliance on a single parallel corpus could be addressed by incorporating multilingual datasets or leveraging semi-supervised techniques to improve cross-lingual generalization. Second, exploring larger pre-trained models (e.g., NLLB-3.3B) as a base for fine-tuning could capture more complex linguistic phenomena and further boost translation quality. Finally, a more nuanced human evaluation is needed to assess subtle aspects of fluency and cultural appropriateness that automated metrics may miss. Pursuing these directions will advance robust machine translation for low-resource languages and promote greater linguistic inclusion.

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
