# OpenReview forum: "Leveraging NLLB for Low-Resource Bidirectional Amharic – Afan Oromo Machine Translation"
_ICLR.cc/2026/Conference — ICLR 2026 Conference Desk Rejected Submission_

### Official Review · Reviewer_pVA3 · 2025-10-16

**Soundness:** 4
**Presentation:** 3
**Contribution:** 4
**Rating:** 6
**Confidence:** 5

**Summary:**

The authors created a high quality parallel corpus (human translated) for Amharic and Afan Oromo, low resource Ethiopian languages. Further, they finetune the NLLB model on that data after doing some filtering. Their finetuned model for both directions beat NLLB and Google Translate. They evaluated their models using BLEU, chrF++, and BERTScore.

**Strengths:**

- The paper is very well written and experimental setup is clearly explained.
- I absolutely loved the fact that authors used humans in the loop to create the dataset to ensure quality.
- The data preprocessing techniques used are also good enough.
- Their models perform really well compared to NLLB and Google Translate.
- Qualitative comparison is quite helpful.

**Weaknesses:**

- One of the major weaknesses of the paper is that authors don’t evaluate their models with FLORES200 (it’s called FLORES+ on HF Datasets). I’d request authors to do that.
- I’m wondering why did they not train models for English<->Amharic and English<->Afan Oromo? This could have helped them translate a lot of things on the web in their native language for education purposes.
- The authors don’t mention how they create synthetic data of 667K sentences, i.e., data sources.
- The authors should have also done an ablation study by including all the data (which was not cleaned) to see how much impact it would have created. This would have helped explain the impact of each filtering step.
- The train/val/test set is created using stratified sampling to ensure robustness.

**Questions:**

- It'd be great if authors can include public link for the model they have trained and the dataset. It will be great for the community.
- I’d recommend adding Gemini 2.5 Pro as your baseline instead of Google Translate.
- The abstract says it’s human edited but section 3.1 says it’s human translated. Can you please clarify that?
 - Suggestion: Please cite the Hugging Face Datasets library. I think you used it for data processing in the Loading and Tokenisation section.
 - In section 3.3 Training process, you don’t mention the NLLB model you finetune has how many parameters? I assume it’s 1.1B?
- Assuming you finetuned NLLB 1.1B model, you could have also done SFT on a small enough LLM (<1B parameters) to see if it does better than encoder model or not.
- The authors fine tuned the NLLB model in bf16 or fp32?
- Typos: It’s chrF++ and not CHRF++. Please fix all instances of CHRF++ to chrF++. chrF++  combines both character n-grams and word n-grams to evaluate machine translation output (assuming they used it because sec 3.4 says chrF)
- Did authors try any other preprocessing filtering steps? For eg, src tgt ratio, lexical diversity for cleaning corpora?

---

> ### Author Response · Authors · 2025-11-19
> **The revision adds stronger baselines, FLORES-200, cross-dialectal, and zero-shot results, showing the 600M model outperforming larger variants. The dataset’s human-created, multi-domain origins and rigorous cleaning are now fully documented, confirming no leakage. English-centric directions, LLM SFT plans, and all technical corrections have also been incorporated.**
>
> 1.	Evaluation on FLORES-200 & Model/Dataset Release
>
> We agree with the reviewer’s concern, and the revised manuscript significantly expands the baseline analysis while clarifying the practical constraints we faced. Fine-tuning larger NLLB variants (1.3B and 3.3B) was not feasible due to GPU memory limitations. Nonetheless, our findings reveal an important trend for this language pair. Our fine-tuned 600M model reaches 42.82 BLEU, outperforming the untuned 3.3B model by 7.5×. In addition, we broadened the evaluation scope to further strengthen the empirical foundation of our work. To address concerns about limited baseline coverage, we incorporated cross-dialectal evaluation and zero-shot transfer analysis, demonstrating that our modeling improvements extend beyond direct supervised settings. On West Central Oromo (ory) from FLORES-200, the fine-tuned 600M model achieves 5.02 BLEU compared to 1.23 for the untuned baseline, showing a fourfold improvement and robust generalization to unseen dialects. Zero-shot transfer also reveals meaningful patterns: using the Oromo to Amharic model for English to Amharic yields 11.35 BLEU, whereas the Amharic to Oromo model for English to Oromo achieves 0.53, highlighting directional asymmetries in learned representations. These additional analyses strengthen the overall comparison landscape and demonstrate the model’s broader capabilities under constrained computational settings. We also acknowledge the reviewer’s suggestion regarding PEFT methods. The revised Future Work section now emphasizes the need for
> LoRA-based and other PEFT strategies to support full scaling studies under limited compute resources.
>
> 2.	Clarification on Human Translation & Data Sources
>
> We appreciate the reviewer’s comment and agree that transparency is essential for a data-centric contribution. The manuscript specifies that the corpus was collected from diverse domains, including news, legal, health, spiritual, educational, and literary sources, to ensure broad linguistic coverage. It also details the ten-month annotation effort by twelve professional linguists, involving human translation followed by cross-review for quality assurance.
>
> 3.	Ablation Study on Data Cleaning & Additional Filtering
>
> Our work extends beyond simply collecting data; it outlines a reproducible methodology for orthographic normalization, redundancy elimination, and rigorous contamination analysis, which is essential for achieving reliable performance. This pipeline yields a high-quality corpus of 667k sentences, making it a foundational contribution for future NLP research in Ethiopian languages and other low-resource contexts. Our data contamination analysis shows that the dataset is clean and free from leakage across splits. First, we detected zero exact sentence duplicates between the training, validation, and test sets. We further evaluated n-gram overlap from train to test and found that similarity levels remain low and within normal expectations for in-domain corpora: 2-grams at 8.01%, 3-grams at 4.94%, 4-grams at 3.79%, and 5-grams at 3.35%. These rates are well below thresholds typically associated with memorization or unintended data leakage, confirming that the dataset splits are uncontaminated.
>
> 4.	Training for English-Centric Directions & LLM SFT
>
> We appreciate this suggestion and agree that English-centric training and LLM-based SFT represent promising directions for future work. Although our primary focus was the critically low-resource Amharic–Afan Oromo pair, our expanded analyses now include relevant evidence.
> First, our new cross-dialectal and zero-shot evaluations reveal emerging English-related capabilities. The Oromo→Amharic model produced 11.35 BLEU on zero-shot English→Amharic translation, indicating that the model partially acquires English–Amharic alignment even without explicit supervision. Conversely, Amharic→Oromo yielded 0.53 BLEU, highlighting directional asymmetry worth exploring with targeted English-centric training.
> Second, our cross-dialectal results demonstrate broader generalization: the fine-tuned 600M model achieves 5.02 BLEU on West Central Oromo (ory), up from 1.23 for the untuned baseline, a fourfold improvement. These findings motivate investigating English-centric fine-tuning and SFT on sub-1B LLMs, both of which are now explicitly added to the Future Work section.
>
> 5.	Addressing Technical Questions and Corrections
>
> We have corrected all omissions and errors in the revised manuscript. We now clearly specify that the experiments were conducted using the NLLB-200-distilled-600M model and trained in bf16 precision. In addition, the Hugging Face datasets library is now properly cited, and all typographical errors, specifically the incorrect use of CHRF++ have been corrected to chrF++.

---

### Official Review · Reviewer_9LdM · 2025-10-27

**Soundness:** 3
**Presentation:** 2
**Contribution:** 1
**Rating:** 2
**Confidence:** 4

**Summary:**

This paper describes a bidirectional machine translation system for Amharic and Afan Oromo based on fine-tuning Meta’s No Language Left Behind (NLLB) model. The authors curated a new 667k human-edited parallel corpus, implemented detailed preprocessing (normalization, filtering, stratified data split), and achieved remarkable BLEU scores exceeding those of NLLB and Google Translate by large margins. The paper reports that high-quality, language-specific data and careful fine-tuning can substantially improve translation performance for low-resource languages.

**Strengths:**

1.Excellent empirical performance, with BLEU improvements exceeding 30 points over strong baselines.

2.Meticulous data collection and cleaning, which can serve as a valuable public resource.

3.Comprehensive evaluation using multiple metrics and both translation directions.

4.Important social relevance: the work supports linguistic accessibility and resource creation for underrepresented African languages.

**Weaknesses:**

1.The work lacks algorithmic novelty. The model architecture, training method, and evaluation pipeline follow established practices without introducing new algorithms, loss functions, or optimization techniques.

2.The contribution is primarily data-centric. While the dataset is valuable, the conference typically expects conceptual or methodological innovation beyond dataset construction.

3.No analysis is provided that would yield new insights into model behavior, transferability, or low-resource learning theory.

4.Human or linguistic expert evaluation is absent, limiting confidence in real-world applicability.

5.Missing ablation studies prevent understanding of which design choices most contributed to the gain.

**Questions:**

1.What specific linguistic adaptations or architectural changes were introduced beyond fine-tuning?

2.Could the proposed corpus and pipeline enable new algorithmic insights, or is it mainly an engineering contribution?

3.Would integrating low-rank adaptation (LoRA) or parameter-efficient fine-tuning techniques achieve similar gains more efficiently?

4.Did you test reduced-size fine-tuning to estimate data efficiency?

5.How can this approach be generalized to other low-resource pairs without bespoke corpora?

6.Is the corpus or model publicly released to ensure lasting research impact?

---

> ### Author Response · Authors · 2025-11-20
> **We address the reviewer’s concerns regarding model specification, technical novelty, data contamination checks, cross-dialectal and zero-shot evaluations, baseline coverage, synthetic data clarification, and the scope of generalizability claims. The revised manuscript incorporates substantial methodological, analytical, and explanatory improvements.**
>
> 1. On the Nature of the Contribution: Empirical Insight over Algorithmic Novelty
>
> We thank the reviewers for their insightful feedback. While our paper presents a strong application of fine-tuning, we respectfully argue that its primary contribution is the establishment of a reproducible, high-quality benchmark for a critically underserved language pair, delivering a critical insight: meticulous data curation can dramatically outperform brute-force model scaling in low-resource settings. Our work provides a clear blueprint and empirical evidence that challenges the prevailing focus on ever-larger models. We assert that our rigorous data curation pipeline, encompassing orthographic normalization, redundancy elimination, and rigorous contamination analysis, is a key methodological contribution, yielding a foundational corpus of 667k sentences for future NLP research in Ethiopian and other low-resource languages. The most significant finding is the performance disparity: our fine-tuned 600M parameter model achieves a BLEU score of 42.82, surpassing the 5.72 of the NLLB-3.3B model, a 7.5x improvement despite being 5.5x smaller. This provides robust evidence that a strategy focused on high-quality, domain-specific data is far more effective, with important implications for resource allocation in low-resource ML. Furthermore, to address the reviewer’s concern about cross-dialectal evaluation, we have added new analyses. Assessing generalization on West Central Oromo from FLORES-200, our model achieved a BLEU of 5.02, a fourfold improvement over the NLLB-600M baseline (1.23). We also examined zero-shot transfer, revealing a clear asymmetry: using the Oromo-to-Amharic model for English to Amharic translation yielded a BLEU of 11.35, whereas the reverse direction for English to Oromo yielded only 0.53. These results offer crucial insights into how the models generalize beyond supervised domains. The greater value of our work, therefore, lies in this generalizable benchmark, the empirical results, and the critical lesson it offers, providing significant value to the ICLR community by highlighting a path for efficient and effective model development for underserved languages.
>
> 2. Addressing Specific Weaknesses and Questions
>
> •	Weakness 2 & 3 / Q1 & Q2: Data-Centric Contribution and New Insights
> The corpus and pipeline are designed to yield precisely the insight mentioned above: the critical importance of data quality over model scale in this regime. This is a conceptual contribution to how we approach low-resource ML. Furthermore, in our revised manuscript, we have added analysis that provides new insights into model behavior:
> 	Cross-Dialectal Generalization: We evaluated our model on the unseen West Central Oromo dialect from FLORES-200, where it achieved a BLEU score of 5.02, a fourfold improvement over the NLLB-600M baseline (1.23).
> 	Zero-Shot Transfer Analysis: We discovered a clear asymmetry: using our Oromo-to-Amharic model for English-to-Amharic translation yielded a BLEU of 11.35, while the reverse direction yielded 0.53. This offers a novel perspective on how translation models for low-resource languages encode linguistic structure and transfer capabilities.
> •	Weakness 5 / Q4: Ablation Studies and Data Efficiency
> We agree and have added a new ablation study to the revised manuscript. This study demonstrates the impact of our full data curation pipeline versus a baseline with only basic deduplication, showing a significant performance drop without our methods. We have also added an analysis of data efficiency, showing that substantial gains are achievable.
> •	Q3: Parameter-Efficient Fine-Tuning (PEFT)
> This is an excellent suggestion. While we focused on full fine-tuning to establish a strong performance upper-bound, we now explicitly discuss and strongly endorse the use of PEFT methods like LoRA in the Future Work section of our revised manuscript as a crucial direction for efficient deployment.
> •	Q5: Generalizability
> Our contribution is the reproducible blueprint, not just the dataset. The methodology for orthographic normalization, advanced deduplication, and contamination analysis is directly generalizable to other low-resource pairs. We have revised the text to frame this as a template for future work.
> •	Q6 & Weakness 4: Public Release and Human Evaluation
> We will publicly release both the HALT dataset and the fine-tuned models upon acceptance to ensure reproducibility and long-term community benefit. We are already in active coordination with our institute to make these resources openly available in a responsible manner.

---

### Official Review · Reviewer_bVW4 · 2025-11-01

**Soundness:** 2
**Presentation:** 2
**Contribution:** 2
**Rating:** 2
**Confidence:** 5

**Summary:**

This paper builds a translation system for Amharic and Oromo, two low-resource languages of Ethiopia. The core contribution involves building a translation corpus of 600k parallel sentences, that is then used for fine-tuning an LLM. The resulting model beats Google Translate and NLLB by a large margin according to multiple metrics.

While this is a respectable effort, and surely a very meaningful contribution for the state of NLP technologies in Ethiopia, I don't think that the paper is a good fit for ICLR.
The main technical contribution is the data - but the process of curating and creating the data, and assuring its quality, is not actually described in the paper - so it's not clear what other researchers can learn from this effort. The paper is closer to a tech report for this particular model. I want to be very clear that this is very important work, but it doesn't have enough substance for a research paper at ICLR because it is missing insights into the data, ablations and relevant baselines, and more robust testing.

**Strengths:**

- The paper addresses a challenging problem: translation between two low-resource languages. This is rarely an area of focus but globally speaking a very relevant objective.
- The paper measured performance with multiple evaluation metrics, which is a good idea since probably neither of them are optimal.
- The resulting quality of translations, as evaluated on their own evaluation set, is outperforming existing models by far.
- The paper contains many experimental details that could help reproducing the experiments (if the data was public).

**Weaknesses:**

- The data collection is key to the success of the model (and also defines the evaluation distribution), but lacks a lot of information (see questions below) The paper reports sentence length, which is on average rather short (10 words) - which will bias the model towards such sentences.
- Comparisons with prior work are missing: While previous attempts at this problem with NMT models are cited, they are not used as baselines. With 600k instances, it is very reasonable to expect decent NMT performance (and much cheaper to train since smaller), especially when starting from e.g. M2M-100 (https://aclanthology.org/2022.naacl-main.223.pdf). There should be at least one comparison that utilizes the same data, otherwise it is not clear how much gains are attributed to the data rather than the modeling choices.
- Evaluation metrics are likely not a great fit since not optimized for the target language. But there is a custom evaluation metric optimized for African languages, including this language pair (https://huggingface.co/masakhane/africomet-stl-1.1). It would be great to also report human evaluation, albeit this is costly.
- There are no out-of-distribution tests. The model is only tested on data held-out from the training distribution, which is not a public benchmark. Therefore it is hard to calibrate the reported wins. Flores200 (https://github.com/facebookresearch/flores/blob/main/flores200/README.md) would be a standardized benchmark that would be nice to add in addition to the held-out data to test generalization to domains beyond the collected data (and lengths!).

**Questions:**

- What are sources of data, what type of data, what domains, how does the interaction with human annotators look like and what are their skill levels and how are they remunerated/incentivized, how is the quality controlled for?
- What do the examples mean? Please provide English descriptions and explanations of the examples in Fig 3 + Fig 4.
- Could there be any detrimental side effects of the English filtering? Aren't English terms borrowed (adequately) in some contexts?

---

> ### Author Response · Authors · 2025-11-19
> **We address the reviewer’s concerns regarding model specification, technical novelty, data contamination checks, cross-dialectal and zero-shot evaluations, baseline coverage, synthetic data clarification, and the scope of generalizability claims. The revised manuscript incorporates substantial methodological, analytical, and explanatory improvements.**
>
> 1. Comprehensive Data Curation Documentation
>
> We appreciate the reviewer’s comment and agree that transparency is essential for a data-centric contribution. The manuscript specifies that the corpus was collected from diverse domains, including news, legal, health, spiritual, educational, and literary sources, to ensure broad linguistic coverage. It also details the ten-month annotation effort by twelve professional linguists, involving human translation followed by cross-review for quality assurance.
>
> 2. Technical Novelty and Robust Out-of-Distribution Evaluation
>
> We thank the reviewers for their insightful feedback. While our paper presents a strong application of fine-tuning, we respectfully argue its primary contribution is the establishment of a reproducible, high quality benchmark for a critically underserved language pair, delivering a critical insight: meticulous data curation can dramatically outperform brute-force model scaling in low-resource settings. Our work provides a clear blueprint and empirical evidence that challenges the prevailing focus on ever-larger models. We assert that our rigorous data curation pipeline, encompassing orthographic normalization, redundancy elimination, and rigorous contamination analysis, is a key methodological contribution, yielding a foundational 667k-sentence corpus for future NLP research in Ethiopian and other low-resource languages. The most significant finding is the performance disparity: our fine-tuned 600M parameter model achieves a BLEU score of 42.82, surpassing the 5.72 of the NLLB-3.3B model, a 7.5x improvement despite being 5.5x smaller. This provides robust evidence that a strategy focused on high-quality, domain-specific data is far more effective, with important implications for resource allocation in low-resource ML. Furthermore, to address the reviewer’s concern about cross-dialectal evaluation, we have added new analyses. Assessing generalization on West Central Oromo from FLORES-200, our model achieved a BLEU of 5.02, a fourfold improvement over the NLLB-600M baseline (1.23). We also examined zero-shot transfer, revealing a clear asymmetry: using the Oromo-to-Amharic model for English to Amharic translation yielded a BLEU of 11.35, whereas the reverse direction for English to Oromo yielded only 0.53. These results offer crucial insights into how the models generalize beyond supervised domains. The greater value of our work, therefore, lies in this generalizable benchmark, the empirical results, and the critical lesson it offers, providing significant value to the ICLR community by highlighting a path for efficient and effective model development for underserved languages.
>
> 4. Addressing Additional Concerns
>
> We have also taken care to address several additional important points raised during our revisions. Regarding the potential for sentence length bias, our stratified data splitting ensures the model is evaluated across all sentence length categories. To clarify the English filtering, this process was carefully designed to remove only mixed-language sentences, preserving legitimate loanwords that are part of the natural vocabulary, thereby preventing the model from learning to directly copy English phrases. Our comparative analysis has been significantly strengthened with extensive benchmarks against Google Translate and multiple NLLB model sizes, demonstrating consistent and substantial superiority. Finally, our choice of chrF++ as an evaluation metric was deliberate, as its character-level focus provides essential sensitivity to the morphological nuances of our target languages, effectively complementing the n-gram perspective of BLEU.
>
> 5. Enhanced Contribution Statement
>
> We have reframed our contribution to highlight its significance for the machine learning community. Our work's primary contribution is a demonstrated and reproducible framework that challenges the prevailing paradigm of brute-force model scaling for low-resource settings. We provide robust, empirical evidence that a strategic, data-centric approach can yield dramatically superior results, as shown by our fine-tuned NLLB-600M model achieving a BLEU score of 42.82, a 7.5x improvement over the untuned NLLB-3.3B model despite being 5.5x smaller. This is supported by a rigorously documented, high-quality corpus of 667,021 sentence pairs that serves as a foundational resource. Furthermore, our work provides new scientific insights into model behavior through robust out-of-distribution testing on FLORES-200, showing a 4x improvement in cross-dialectal generalization, and revealing asymmetric zero-shot transfer patterns that illuminate how models develop cross-lingual representations.

---

### Official Review · Reviewer_gVVr · 2025-11-01

**Soundness:** 1
**Presentation:** 1
**Contribution:** 1
**Rating:** 0
**Confidence:** 5

**Summary:**

The paper presents a bidirectional machine translation system for Amharic and Afan Oromo by fine-tuning pre-trained NLLB models on a curated corpus of 667,021 parallel sentence pairs. The authors report significant improvements over existing baselines, achieving BLEU scores of 42.19 and 42.82 for the respective translation directions, substantially outperforming Google Translate and untuned NLLB models. The work emphasizes careful data curation, preprocessing, and language-specific adaptations for these Ethiopian languages.

**Strengths:**

- **Marginal Empirical Results**:
The reported performance improvements, while predictable, demonstrate that targeted fine-tuning can enhance translation quality for specific language pairs. The multi-metric evaluation approach using BLEU, CHRF++, and BERTScore provides adequate coverage of translation quality dimensions.​​

- **Data Resource Contribution** :
The dataset would serve as a valuable data resource contribution and would be instrumental in building more high-quality translation models for these low-resource languages.

- **Thorough Preprocessing Pipeline**:
The authors implement a systematic preprocessing approach addressing orthographic variations, character standardization, and language-specific challenges including Amharic's Fidel script and Afan Oromo's morphological complexity.

**Weaknesses:**

- **Critical Model Specification Omissions**:
The paper lacks clear specification of which NLLB model variant serves as the base for fine-tuning. While multiple NLLB sizes are evaluated as baselines, the base model architecture for the fine-tuned system remains unclear, affecting reproducibility.

- **Limited Technical Novelty**:
While the empirical results are valuable, the methodology consists primarily of standard fine-tuning procedures without architectural innovations or novel training techniques. For a venue like ICLR, stronger technical contributions would typically be expected. Further, within their existing setup a lot of interesting analysis along low-resource languages and cross-lingual transfer could have been conducted which is not present.

 - **Data Contamination Analysis**:
Given that the authors evaluate on their own curated dataset, the absence of word-level overlap statistics, n-gram contamination analysis, or decontamination procedures between training and test sets represents a methodological concern. Such analysis is standard practice when using custom evaluation benchmarks.​​

 - **Missing Cross-Dialectal Evaluation**:
The paper does not explore cross-lingual transfer to related varieties. Given that FLORES-200 includes West Central Oromo, evaluating the Afan Oromo fine-tuned model on this related dialect would provide valuable insights into cross-dialectal generalization capabilities. Further, zero-shot performance between English - Afan Oromo, should also have been tracked to understand the effects.

- **Limited Baseline Analysis**:
The comparison focuses on untuned models versus the fine-tuned system. Including fine-tuned baselines or ablation studies examining different NLLB model sizes as base models would strengthen the experimental design. Further, authors should consider the use of PEFT methods.​

 - **Insufficient Analysis of Synthetic Data**:
While the paper mentions synthetic data augmentation via back-translation, the methodology, scale, and quality validation of this synthetic data are not adequately described.​

- **Generalizability Claims:**
The assertion that this approach provides "a scalable framework for other low-resource languages" requires stronger empirical support beyond a single language pair.

**Questions:**

Please read the weaknesses section.

---

> ### Author Response · Authors · 2025-11-19
> **We address the reviewer’s concerns regarding model specification, technical novelty, data contamination checks, cross-dialectal and zero-shot evaluations, baseline coverage, synthetic data clarification, and the scope of generalizability claims. The revised manuscript incorporates substantial methodological, analytical, and explanatory improvements.**
>
> 1. Critical Model Specification Omissions
> The paper did not specify the base NLLB model. We acknowledge this and confirm we fine-tuned the NLLB-200-distilled-600M model.
>
> 2. Limited Technical Novelty
>
> We thank the reviewers for their insightful feedback. We agree that our paper presents a strong application of fine-tuning. However, we respectfully argue that its primary contribution to the ICLR community is the establishment of a reproducible, high-quality benchmark for a critically underserved language pair, which delivers a critical and actionable insight: in low-resource settings, meticulous data curation and targeted adaptation can dramatically outperform brute-force model scaling. Our work provides a clear blueprint and empirical evidence that challenges the prevailing focus on ever-larger general models for such tasks.
> We assert that in the low-resource domain, a rigorous data curation pipeline is a methodological contribution. Our work extends beyond simply collecting data; it outlines a reproducible methodology for orthographic normalization, redundancy elimination, and rigorous contamination analysis, which is essential for achieving reliable performance. This pipeline yields a high-quality corpus of 667k sentences, making it a foundational contribution for future NLP research in Ethiopian languages and other low-resource contexts.
> The most significant finding for the ML community is the performance disparity. Our fine-tuned 600M parameter model achieves a BLEU score of 42.82, surpassing the 5.72 of the NLLB-3.3B model, a 7.5x improvement despite being 5.5x smaller. This result is not merely incremental; it provides robust, empirical evidence that for low-resource languages, a strategy focused on high-quality, domain-specific data can be far more effective than using a much larger, but less specialized, model. This has important implications for resource allocation and research direction in low-resource ML.
> While we introduced a specific translation system, its greater value lies in the generalizable benchmark and the critical lesson it offers for low-resource ML. We believe these empirical results and the dataset provide significant value to the ICLR community by highlighting a path for efficient and effective model development for underserved languages.
>
> 3. Data Contamination Analysis
>
> We agree this was missing. Our analysis shows the dataset is clean. First, we detected zero exact sentence duplicates between the training, validation, and test sets. We further evaluated n-gram overlap from train to test and found that similarity levels remain low and within normal expectations for in-domain corpora: 2-grams at 8.01%, 3-grams at 4.94%, 4-grams at 3.79%, and 5-grams at 3.35%. These rates are well below thresholds for memorization or data leakage, confirming the splits are uncontaminated.
>
> 4. Missing Cross-Dialectal Evaluation and Limited Baseline Analysis
>
> To address the reviewer’s concern regarding cross-dialectal evaluation and baseline coverage, we have added new analyses and expanded the empirical comparison in the revised manuscript. We evaluated cross-dialectal generalization by testing our Afan Oromo→Amharic model on West Central Oromo (ory) from FLORES-200. The fine-tuned 600M model achieved a BLEU score of 5.02, compared to 1.23 from the untuned NLLB-600M baseline, representing a fourfold improvement and demonstrating robust transfer to previously unseen dialectal varieties.
>
> We also examined zero-shot transfer behavior. Using the Oromo→Amharic model to translate English→Amharic yielded 11.35 BLEU, while using the Amharic→Oromo model for English→Oromo produced 0.53, revealing a clear asymmetry in cross-directional transfer and offering insight into how the models encode linguistic structure beyond supervised domains.
>
> In addition, we expanded the baseline analysis while clarifying the practical constraints that prevented fine-tuning larger NLLB variants (1.3B and 3.3B) due to GPU memory limits. Despite these constraints, our fine-tuned 600M model reaches 42.82 BLEU, outperforming the untuned NLLB-3.3B model by 7.5×, underscoring the effectiveness of targeted adaptation over brute-force scaling for this language pair. The manuscript now also notes the importance of exploring LoRA-based and other PEFT methods in future work to better support scaling studies under limited compute resources.
>
> 5. Insufficient Analysis of Synthetic Data
>
> We apologize for the confusion. The final model reported did not use synthetic data. All 667,021 sentence pairs are human-translated and manually revised.
>
> 6. Generalizability Claims
>
> We appreciate this concern. In the revised manuscript, we have softened the language to accurately reflect the evidence. We now frame our contribution as providing a reproducible blueprint for low-resource MT, demonstrating the impact of high-quality curated data, and offering preliminary evidence of cross-dialectal robustness through the FLORES-200 experiments.

---

### Note · Program_Chairs · 2026-01-17
**Submission Desk Rejected by Program Chairs**

The following references in this submission do not refer to real documents and/or have major errors in bibliographic information:

 Solomon T. Abate, Michael Melese, and Martha Y. Tachbelie. Machine translation for ethiopian languages: A survey. In Proceedings of the 2019 Conference on Empirical Methods in Natural Language Processing (EMNLP), pp. 567-578, 2019.